# Distant Recurrence of a Cerebral Cavernous Malformation in the Vicinity of a Developmental Venous Anomaly: Case Report of Local Oxy-Inflammatory Events

**DOI:** 10.3390/ijms232314643

**Published:** 2022-11-24

**Authors:** Andrea Bianconi, Luca Francesco Salvati, Andrea Perrelli, Chiara Ferraris, Armando Massara, Massimiliano Minardi, Gelsomina Aruta, Miriam Rosso, Barbara Massa Micon, Diego Garbossa, Saverio Francesco Retta

**Affiliations:** 1Division of Neurosurgery, Department of Neurosciences “Rita Levi Montalcini”, City of Health and Science and University of Turin, 10124 Torino, Italy; 2CCM Italia Research Network, National Coordination Center, Department of Clinical and Biological Sciences, University of Turin, 10124 Orbassano, Italy; 3Division of Neurosurgery, Santa Corona Hospital, 17027 Pietra Ligure, Italy; 4Department of Clinical and Biological Sciences, School of Medicine and Surgery, University of Turin, Regione Gonzole 10, 10124 Orbassano, Italy; 5Department of Pharmacology and Physiology, University of Rochester Medical Center, Rochester, NY 14602, USA

**Keywords:** cerebrovascular diseases, cerebral cavernous malformation (CCM), developmental venous anomaly (DVA), oxidative stress, sterile inflammation

## Abstract

Background: Cerebral cavernous malformations (CCMs) are a major type of cerebrovascular lesions of proven genetic origin that occur in either sporadic (sCCM) or familial (fCCM) forms, the latter being inherited as an autosomal dominant condition linked to loss-of-function mutations in three known CCM genes. In contrast to fCCMs, sCCMs are rarely linked to mutations in CCM genes and are instead commonly and peculiarly associated with developmental venous anomalies (DVAs), suggesting distinct origins and common pathogenic mechanisms. Case report: A hemorrhagic sCCM in the right frontal lobe of the brain was surgically excised from a symptomatic 3 year old patient, preserving intact and pervious the associated DVA. MRI follow-up examination performed periodically up to 15 years after neurosurgery intervention demonstrated complete removal of the CCM lesion and no residual or relapse signs. However, 18 years after surgery, the patient experienced acute episodes of paresthesia due to a distant recurrence of a new hemorrhagic CCM lesion located within the same area as the previous one. A new surgical intervention was, therefore, necessary, which was again limited to the CCM without affecting the pre-existing DVA. Subsequent follow-up examination by contrast-enhanced MRI evidenced a persistent pattern of signal-intensity abnormalities in the bed of the DVA, including hyperintense gliotic areas, suggesting chronic inflammatory conditions. Conclusions: This case report highlights the possibility of long-term distant recurrence of hemorrhagic sCCMs associated with a DVA, suggesting that such recurrence is secondary to focal sterile inflammatory conditions generated by the DVA.

## 1. Introduction

Cerebral cavernous malformations (CCMs), also known as cavernomas or cavernous angiomas, are vascular malformations characterized by clusters of abnormally dilated capillary channels lined by a thin and leaky endothelial cell layer predisposed to rupture [1,2]. They are the second most common vascular malformation of the central nervous system (CNS) after developmental venous anomalies (DVAs), and they are most sensitively diagnosed by magnetic resonance imaging (MRI), often showing a peculiar popcorn-like appearance and a peripheral rim of hemosiderin [3]. Their prevalence is estimated to be around 0.4–0.5% in general population, although they manifest symptomatically in only 30% of cases [2,4]. CCM is a genetically determined disease, which may arise in a familial form inherited as an autosomal dominant condition with incomplete penetrance and highly variable expressivity, referred to as familial CCM disease (fCCM); however, it most frequently occurs sporadically (sCCM) [5,6]. Specifically, the familial and sporadic forms account for approximately 20% and 80% of the CCM cases, respectively. So far, fCCM has been associated with loss-of-function mutations in three genes, *KRIT1*/*CCM1*, *CCM2* and *PDCD10*/*CCM3*, which have been implicated in a plethora of essential physiological functions, including modulation of cadherin- and integrin-mediated cell adhesion, actin cytoskeleton dynamics, and signal transduction involved in the maintenance of endothelial barrier stability [7]. However, no mutation in these CCM genes were found in up to 15% of fCCM cases, suggesting the potential existence of additional causative determinants [8,9]. Conversely, mutations in the three known CCM genes were rarely detected in sCCM lesions, which instead resulted associated mainly with somatic mutations in other genes [10,11,12], suggesting that fCCM and sCCM lesions may have a distinct genetic origin. Moreover, patients with fCCM often present multiple cerebral cavernomas (up to several hundreds), which are typically not associated with a DVA [13], while patients with sCCM commonly present a single lesion of various sizes (from mm to several cm) that is frequently associated with a DVA, suggesting the possibility of a different developmental mechanism [13,14,15,16,17].

DVAs are the most common cerebrovascular malformations, accounting for up to 60% of the total and being present in approximately 3% of the general population [18,19]. They are congenital malformations that are incidentally or routinely detected by both contrast-enhanced computed tomography (CT) and MRI scans of the brain, and they usually consist of several radially arranged veins that drain centrally into a single dilatated venous trunk, assuming the classical radiological appearance of a caput medusae sign, also known as a palm tree sign [20]. Histologically, the veins are enlarged and sometimes hyalinized, but otherwise normal. The frequent coexistence of sporadic CCMs and DVAs has long been known, and there is evidence that they occur in up to 44% of CCM patients [13,16,21,22].

CCMs can either remain asymptomatic or manifest across a wide range of different clinical symptoms, including seizure (30% of cases), intracranial hemorrhage (ICH) (26% of cases), and focal neurological deficits (16% of cases) [23], with an annual hemorrhage rate estimated between 0.4% and 2.5% [23,24,25]. To date, there are no direct therapeutic approaches for CCM disease other than surgical removal of accessible lesions in symptomatic patients [26,27]. However, while a novel grading for surgical decision making was recently suggested with the aim of improving and standardizing the treatment of CCMs, surgical indications for CCM remain significantly center- and surgeon-dependent [28]. On the other hand, DVAs have always been considered benign lesions, with hemorrhage rates as low as 0.22–0.68% per year, and they usually remain asymptomatic over the entire lifetime [29,30]. The intimate relationship between DVAs and CCMs has been widely studied in the past years, mainly suggesting that sporadic CCMs are secondary to pre-existing DVAs [13,14,31,32,33]. The current most accredited theory of CCM and DVA concurrence is based on the idea that an increased systemic and local venous pressure due to venous congestion and outflow obstruction in DVA carriers leads to recurrent small hemorrhages and brain ischemia, stimulating abnormal angiogenesis [31,34,35,36]. The new fragile and enlarged capillary vessels are susceptible to bleeding, and these small bleeds might significantly contribute to CCM lesion formation. Generally, it has been reported that anatomical angioarchitectural factors of pre-existing DVAs might be the key factors in inducing concurrent sporadic CCMs within the DVA territory by causing disturbance of blood flow and altered hemodynamics [15,34,37,38,39]. In this regard, it is noteworthy that the prevalence of both abnormal angioarchitectural factors of DVAs and DVA-associated CCMs increases with age, suggesting that DVA-associated CCMs are indeed acquired lesions secondary to DVA angioarchitectural changes and consequent alterations in the adjacent brain parenchyma [31,40,41].

The pathophysiology of CCM disease remains to be fully elucidated, as there is still some controversy regarding the multiple mechanisms proposed so far. Among others, recent genetic analyses of surgically excised lesions from sCCM patients and distinct studies in conditional knockout mouse models have clearly demonstrated that homozygous loss-of-function mutations in CCM genes are neither necessary nor sufficient to cause the development of CCM lesions, thereby limiting the two-hit model proposed previously [42,43] and suggesting the implication of additional determinants, including local stressful conditions [8,9]. Consistently, whereas the clinical expression of CCM disease is known to be highly variable even among family members carrying the same CCM gene mutation, microenvironmental stressors, including inflammatory and oxidative stress events, and interindividual variability in stress response have progressively emerged as important determinants of CCM disease pathogenesis and severity [7,44,45,46,47,48,49,50,51]. Furthermore, there is clear evidence that the de novo formation of CCM lesions can occur in sporadic patients with or without a history of treatment with cranial radiation, and they may be more common than currently appreciated [52,53]. In this light, it is, therefore, possible to envisage that chronic microenvironmental stressful conditions generated locally by a DVA might contribute significantly to the formation and progression of an associated CCM lesion.

Herein, we report a case of long-term distant recurrence of a sporadic CCM lesion in the proximity of a voluminous DVA draining in the superior sagittal sinus (SSS), and provide contextualized support to the intriguing hypothesis that such recurrence is secondary to local sterile oxy-inflammatory conditions generated by the DVA.

## 2. Case Description

Our case report concerns a patient affected by the sporadic form of CCM disease, who underwent two neurosurgical interventions, 18 years apart, for the excision of an original and a recurrent symptomatic CCM lesion adjacent to the same pre-existing DVA. Specifically, in the year 2000, at the age of 3, the patient underwent neurosurgery following the diagnosis of intracerebral bleeding caused by a hemorrhagic CCM lesion located in the right frontal lobe of the brain and associated with voluminous DVA draining in the superior sagittal sinus (SSS) (Figure 1). In accordance with guideline recommendations to preserve the DVA in order to avoid serious injuries and adverse consequences [18,54,55,56], the neurosurgical intervention was limited to excision of the CCM lesion, preserving the integrity and patency of the associated DVA, considering also its large size and drainage into the SSS, and the high risk of venous infarction. The diagnosis of CCM disease was subsequently confirmed by histological examination of the surgically resected mass, whereas postoperative MRI examination demonstrated complete removal of the CCM lesion. Then, the patient underwent regular MRI follow-up, showing no residual or relapse signs until the year 2015 (at the age of eighteen), when the follow-up was suspended. No antiepileptic treatment was necessary during the follow-up period.

Three years later, in June 2018, the patient experienced an acute onset of critical episodes of paresthesia in the left lower limb, with ascending progression to the left arm and face. A brain CT scan was, therefore, performed, which revealed a right frontal parasagittal hemorrhagic lesion. Diagnostic completion with MRI showed the presence of a vascular lesion, highly compatible with CCM imaging features, located within the hemorrhagic focus and close to the pre-existing known DVA (Figure 2A).

Neurosurgery was then performed through the same approach used for the first intervention. Briefly, in both the first and the second resections, a right parasagittal craniotomy was performed, without exposing the superior sagittal sinus for safety reason. After dural opening, en bloc resection was achieved in both cases.

During the second neurosurgical intervention, a 1 cm large mulberry-like CCM lesion was detected along the medial hematoma wall next to the falx. The mass was surgically removed en bloc and sent for histological examination, which confirmed the diagnosis of a newly formed CCM lesion. The DVA was preserved intact for the same reasons as the previous surgical intervention. Postoperative MRI examination demonstrated complete removal of the CCM lesion (Figure 2B). The patient’s National Institutes of Health Stroke Scale (NIHSS) scores at admission and discharge were 2 and 0, respectively, and the patient received antiepileptic treatment with levetiracetam 1500 mg/day over the course of hospitalization. The postoperative course was regular, with no additional neurological deficits, and no other recurrence was found during the subsequent follow-up period. However, follow-up examination using gadolinium-based contrast-enhanced brain MRI sequences, including T1-weighted and fluid-attenuated inversion recovery (FLAIR) scans, performed at 1 year after CCM surgery revealed the presence of signal-intensity abnormalities in the bed of the DVA, including point-like contrast enhancements (Figure 3A,C) and hyperintense gliotic areas (Figure 3B,D), suggesting the presence of chronic oxy-inflammatory conditions. While additional imaging approaches, including positron emission tomography (PET) scans and specific molecular analyses, are required to confirm their occurrence, our preliminary observations raise the possibility that such focal oxy-inflammatory conditions contribute to the formation and recurrence of DVA-associated sCCM lesions, thereby providing a useful framework for future investigations in a large number of DVA-associated sCCM cases.

## 3. Discussion

This case report demonstrated the recurrence of a sporadic cerebral cavernous malformation (CCM) in the vicinity of a voluminous developmental venous anomaly (DVA) 18 years after neurosurgical excision of the first CCM lesion, and it provided preliminary neuroimaging data suggesting that such recurrence is secondary to focal oxy-inflammatory conditions generated by the pre-existing DVA. Further circumstantial evidence that supports this possibility is provided and discussed in detail below.

### 3.1. A Unifying Pathogenic Scenario for the Distinctive Features of sCCM and fCCM Lesions

CCMs are abnormally dilated and angiographically occult capillary sinusoids, whose natural history investigation was effectively possible only after the advent of magnetic resonance imaging (MRI), which eventually became the optimal standard tool for their clinical diagnosis and follow-up, as well as for a better knowledge of their peculiar clinical and epidemiological features [3,57,58]. Indeed, it is now well known that the sporadic (sCCM) and familial (fCCM) forms of CCM disease have a significantly distinct set of properties. Among others, these include a higher prevalence of the sCCM (80%) than fCCM (20%) form in the general population, the presence of single and multiple (up to several hundreds) lesions, respectively, and the highly prevalent association of sCCM lesions with common abnormalities of venous drainage, such as DVAs, which increases with age but occurs very rarely in the case of fCCMs [13,59]. Moreover, recent high-throughput genetic analyses of surgical specimens from sCCM patients have revealed that mutations in CCM genes (*CCM1*/*2*/*3*), known to be responsible for at least 80% of fCCM cases [5,60], are rarely detected in sCCM lesions, which are instead more frequently associated with mutations in other genes, including gain-of-function mutations in genes implicated in the PI3K/Akt and MAPK pathways, such as *PIK3CA* and *MAP3K3*, thereby suggesting a distinct genetic origin [10,11,12]. At the same time, sCCM and fCCM lesions share a typical popcorn-like MRI appearance and identical histopathological features, consisting of abnormally dilated and mulberry-shaped clusters of capillary sinusoids lined by a thin endothelium and embedded in a collagen matrix devoid of other neurovascular unit (NVU) and blood–brain barrier (BBB) components, including astrocyte foot processes and pericytes [61]. Taken together, these considerations suggest that sCCM and fCCM lesions share common pathogenic mechanisms despite being characterized by the largely distinct set of properties mentioned above.

Among the potential pathogenic mechanisms that may underlie the development of both sCCM and fCCM lesions and modulate the severity of major associated symptoms, such as focal intracerebral hemorrhage (ICH) and neurological deficits, there is an increased sensitivity of key NVU components, including endothelial cells and astrocytes, to focal acute or chronic oxidative stress and inflammatory (oxy-inflammatory) conditions [9,45,50]. Several studies in cellular models have in fact demonstrated that loss of function of CCM genes sensitizes cells to oxy-inflammatory insults by affecting redox homeostasis and signaling, consequently exerting pleiotropic downstream effects on key redox-sensitive mechanisms that ensure proper cellular homeostasis and defenses against oxidative stress and inflammation, including transcriptional signaling pathways and autophagy [9,47,48,49,62,63,64,65,66]. In particular, CCM gene loss of function causes an imbalance between production of reactive oxygen species (ROS) and the ability of cellular antioxidant mechanisms to detoxify reactive intermediates or repair oxidative damage, thus leading to an enhanced cell susceptibility to oxidative stress-mediated molecular and cellular dysfunctions. This effect occurs through various redox-dependent mechanisms, including activation of the PI3K/Akt and MAPK signaling pathways [48,62], downregulation of the transcription factor FoxO1 and its downstream targets, such as the antioxidant enzyme superoxide dismutase 2 (SOD2) [62], upregulation of the JNK/c-Jun/COX-2 axis [63] and the NADPH oxidase/ROS/NF-κB cascade [47], and activation of mechanistic target of rapamycin (mTOR) and consequent downregulation of autophagy, an essential cellular antioxidant system [64,67,68]. In turn, the persistent alteration of ROS homeostasis and redox signaling leads to a sustained upregulation of NRF2 (nuclear factor E2-related factor 2), the master transcriptional regulator of antioxidant and cytoprotective responses [48,49], as well as to an increased S-glutathionylation of redox-sensitive proteins [65], which eventually results in abnormal adaptive responses that impair microvessel barrier function and exacerbates vascular permeability triggered by inflammatory stimuli [47,50]. On the other hand, accumulated evidence in both fCCM patients and animal models indicates that mutations in CCM genes are not sufficient to induce CCM lesion formation but require the critical contribution of additional pathogenic determinants, including oxy-inflammatory events generated locally by either endogenous or exogenous factors [7,9,45,69,70]. Accordingly, whereas it is well established that oxidative stress and inflammation are intertwined processes [45,71,72], it is also known that oxy-inflammatory events are a major cause of NVU remodeling and dysfunctions associated with the pathogenesis of cerebrovascular diseases, leading to abnormal angiogenic responses and reduced BBB stability [73]. Specifically, among the potential causative factors there are low fluid shear stress conditions, which have been shown to upregulate key signaling pathways implicated in CCM disease pathogenesis, including transforming growth factor beta (TGF-β), Toll-like receptor 4 (TLR4), and oxidative stress pathways [74]. Then, other relevant endogenous factors are damage-associated molecular patterns (DAMPs) released by damaged tissues or activated glial cells and endothelial cells upon sterile inflammatory conditions, such as atherosclerosis, thrombosis, neurodegeneration, or traumatic brain injury, which are known to trigger TLR4-mediated innate immune-inflammatory and oxidative stress responses [9,75,76]. Moreover, additional sources of oxy-inflammatory events already implicated in CCM disease pathogenesis are pathogen-associated molecular patterns (PAMPs), including bacterial PAMPs known to activate TLR4 signaling, such as lipopolysaccharide (LPS) [69], as well as hypoxia [70]. Furthermore, consistent with the fact that focal oxy-inflammatory conditions contribute significantly to CCM lesion formation and severity, distinct genetic modifiers of oxy-inflammatory responses have been identified as linked to the incomplete penetrance and highly variable expressivity of CCM disease in large cohorts of fCCM cases [8,44,46,51]. In addition, new intriguing and significant evidence in animal models suggests that CCM gene mutations can have pathological effects not limited to CCM disease, as they may predispose to the development of other pathological conditions associated with abnormal oxy-inflammatory responses, such as atherosclerosis and hepatic metabolic, antioxidant, and antiglycative dysfunctions [77,78]. Overall, these and other original discoveries made over the last decade have brought the plethora of signaling pathways and molecular mechanisms so far implicated in CCM disease pathogenesis into a unique and unifying mechanistic scenario, whereby a combination of genetic and microenvironmental determinants of increased susceptibility and sensitivity to focal oxy-inflammatory insults takes the central stage [9,51]. In this new light, it is, therefore, possible to speculate that chronic focal oxy-inflammatory conditions generated by DVAs, either directly or through adjacent activated glial cells, may be sufficient to induce the formation of sCCM lesions, thereby providing a plausible explanation for the higher prevalence of the sporadic form of CCM disease.

### 3.2. DVA: A Chronic Condition That Can Trigger Sterile Inflammation

DVAs are congenital abnormalities of cerebral venous drainage, consisting of multiple small medullary veins that join to drain into a larger collector vein, which then drains superficially into a dural sinus, or less often into the deep venous system. Additionally known as cerebral venous angiomas or cerebral venous malformations, DVAs are the most common vascular malformations that occur in the CNS, and they are easily identified by contrast-enhanced brain MRI or susceptibility weighted imaging (SWI) sequences, assuming a characteristic “caput medusae” appearance [79]. Normal brain tissue is present between the veins, and there is no abnormal shunt. Neuroimaging studies by CT or MRI have often identified various brain parenchymal abnormalities within the drainage territory of DVAs, including regional atrophy, edema, leukoaraiosis, gliosis (as also reported in this study), and dystrophic calcification, which were mainly attributed to altered hemodynamics and venous congestion or hypertension [80]. Specific perfusion studies indeed confirmed the occurrence of altered hemodynamics in the DVA drainage area, including prolonged mean transit time (MTT) and time to maximum of the residue function (Tmax), as well as increased cerebral blood volume (CBV) [38,81,82]. Moreover, signal-intensity abnormalities in the brain parenchyma adjacent to a DVA have been shown to correlate with DVA angioarchitectural factors that cause disturbance of blood flow, including number and tortuosity of medullary veins, and angulation, tortuosity, stenosis, and thrombosis of the draining vein [15,80], as well as to increase with age [41,83]. In addition, distinct functional neuroimaging findings, based on positron emission tomography (PET), have remarkably demonstrated that such signal-intensity abnormalities are frequently related to significant metabolic alterations [84,85,86]. Given the established strong association of both disturbed blood flow and altered metabolism with inflammation [87,88,89,90], it is, therefore, conceivable that the signal-intensity abnormalities that occur in the brain parenchyma adjacent to a DVA may be related to sterile inflammatory conditions caused by the local release of damage-associated inflammatory inducers, such as DAMPs (alarmins) and consequent glial activation [75,76,91]. Consistently, whereas some DVAs are indeed found associated with gliosis, there is emerging evidence that these venous anomalies are implicated in the pathogenesis of major neuroinflammatory diseases, such as multiple sclerosis [92,93].

### 3.3. Focal Coexistence of DVA and sCCM: A Dangerous Liaison Linked to Sterile Inflammation

Accumulated evidence unequivocally indicates that structural and hemodynamic conditions within the DVA are causative factors for the de novo formation of sCCM lesions [14,15,37,94]. Specifically, it has been demonstrated that the same angioarchitectural features that cause brain parenchymal abnormalities within the drainage territory of DVAs, including number and tortuosity of DVA medullary veins, and angulation, tortuosity, stenosis, and thrombosis of the collecting vein, are strongly correlated with the prevalence of DVA-associated sCCM lesions, suggesting a pathogenic link [15,94]. Accordingly, whereas high-resolution brain imaging with 7 T MRI has consistently found abnormal venous drainage in association with sCCMs [94], further support for a causative relationship is provided by clear evidence for the age-dependent increase in the prevalence of both DVA-related signal-intensity abnormalities in the adjacent brain parenchyma and DVA-associated sCCM lesions [40,41,83]. Intriguingly, the fact that DVAs occur with sporadic but not with familial CCMs has suggested the possibility of a different developmental mechanism, whereby sCCMs are mainly induced by specific angioarchitectural and hemodynamic factors typical of local venous anomalies, while fCCMs derive from endothelial dysfunction consequent to a loss-of-function mutation in CCM genes [13,32,94]. On the other hand, along with the fact that sCCM and fCCM lesions share virtually identical clinical, neuroradiological, and histopathological features, the evidence accumulated in cellular and animal models described above points to very common, if not identical, molecular mechanisms of pathogenesis, including a major role for pleiotropic redox-sensitive mechanisms that influence the susceptibility of brain capillaries to focal oxy-inflammatory challenges (Figure 4). Consistently, whereas the tight interplay among oxidative stress, inflammation, and angiogenesis is obviously implicated in the pathogenic effects of DVA-related abnormal angioarchitectural and hemodynamic factors, including the genesis of signal-intensity abnormalities in the adjacent brain parenchyma and the secondary formation of concurrent sCCM lesions, it has also been implicated in the pathological effects of loss-of-function mutations in CCM genes [9,45]. In particular, the possibility for a crucial role of sterile inflammatory mechanisms as main drivers of both sCCM and fCCM lesion formation in genetically susceptible individuals is supported by the original discovery that TLR4-mediated innate immune-inflammatory responses and polymorphisms that increase TLR4 gene expression are major determinants of CCM disease pathogenesis and severity, while inhibition of TLR4 signaling may prevent CCM lesion formation [69]. Moreover, further support is notably provided by a recent report showing that the co-occurrence of infratentorial DVAs and chronic inflammation increases the odds of sCCM lesion formation [32]. Thus, although alternative hypotheses cannot be ruled out, including the DVA hypertension-related ischemic or hemorrhagic angiogenic proliferation hypothesis [15,94], growing evidence points to sterile inflammatory responses as the key causative factors responsible for the secondary formation of DVA-associated sCCM lesions. Moreover, this suggests that inflammatory responses are also crucially implicated in the development and severity of CCM lesions not associated with DVAs, including both sCCMs and fCCMs, in individuals carrying CCM gene mutations and/or other genetic risk factors of susceptibility to focal oxy-inflammatory insults. Intriguingly, consistent with this hypothesis, there is compelling evidence that sterile inflammatory events, including innate immune-inflammatory responses, play a primary role in in triggering and sustaining both oxidative stress and angiogenic responses [45,72,95].

## 4. Concluding Remarks and Implications for Future Research

Overall, on the basis of the compelling evidence described above, it is possible to propose that sterile inflammatory responses induced in the brain parenchyma by changes in angioarchitectural and hemodynamic features of a pre-existing DVA may represent a novel plausible etiological mechanism for the long-term distant recurrence of a DVA-related hemorrhagic sCCM lesion identified in our study. Indeed, in addition to correlating with the age-dependent increase in the prevalence of sCCMs associated with DVAs [40], such a mechanism might also be associated with the age-dependent increase in the prevalence of structural and metabolic signal-intensity abnormalities detected in the brain parenchyma of DVAs using MRI and PET analyses, respectively [41,83,84,85,86]. Furthermore, the fact that focal chronic oxy-inflammatory conditions generated by distinct pathogenic determinants can affect the stability of sensitive capillary vessels located nearby also raises the possibility that this pathological effect influences the onset and severity of both sCCM and fCCM lesions, thereby suggesting that different causative determinants eventually converge into common molecular mechanisms and outcomes.

While specific molecular and pathological analyses are required to confirm the occurrence of DVA-associated sterile inflammatory responses, our observational study provides proof of concept of the presence of neuroimaging signal-intensity abnormalities attributable to neuroinflammatory events, including point-like contrast enhancements and hyperintense gliotic areas in the brain parenchyma surrounding a distant recurrent sCCM lesion associated to a pre-existing large DVA. Accordingly, despite several limitations, our case report paves the way for future comprehensive investigations in a large number of DVA-associated sCCM cases through advanced imaging approaches more suitable for studying brain inflammation, including advanced PET guided neuroinflammation imaging with specific tracers [96,97,98,99]. Furthermore, our observational study implies also that the development of treatment approaches aimed at targeting sterile inflammation might enable control of neuroinflammation and improve CCM patient outcomes. To this regard, it is noteworthy that distinct therapeutic approaches based on nanotechnology indeed represent a promising hope for the treatment of many neuroinflammation-related congenital and acquired disorders, including CCM disease [68,100,101]. Lastly, our case report indicates that a DVA left in place after surgical removal of a CCM lesion may promote the relapse of adjacent CCM lesions, thereby implying that the combined resection of CCM lesions and associated DVAs could reduce the risk of recurrence. Notably, while such implication is in contrast with the main paradigm in cavernoma surgery, whereby DVAs must be preserved in order to avoid adverse consequences [54,55,56], it is in agreement with other authors reporting that resection of both CCM lesions and associated DVAs can reduce the risk of recurrent CCMs [21], suggesting the necessity of a careful reconsideration of current surgical practices in DVA-associated CCM care.

## Figures and Tables

**Figure 1 ijms-23-14643-f001:**
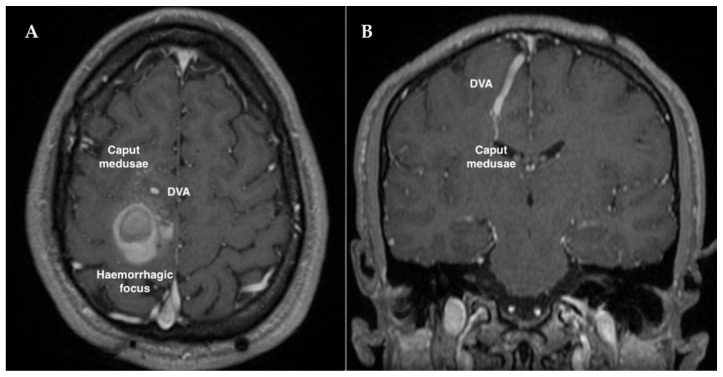
First preoperative T1 contrast-enhanced MRI showing a hemorrhagic cavernoma in the vicinity of a voluminous DVA, with its typical caput medusae sign, in the right frontal lobe: (**A**) axial view; (**B**) coronal view.

**Figure 2 ijms-23-14643-f002:**
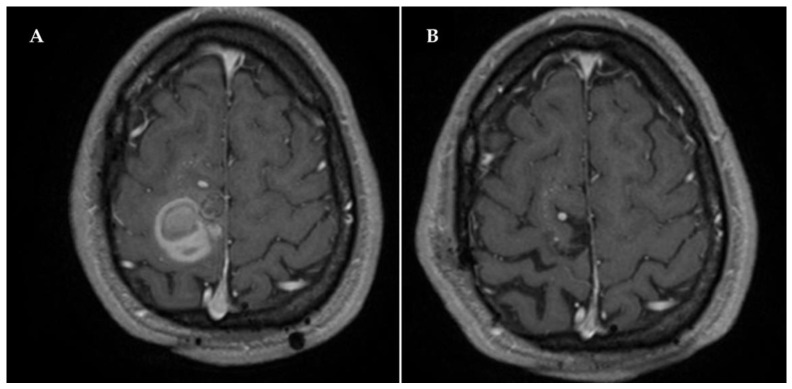
(**A**) Axial MRI view of the patient’s brain, showing the presence of a newly formed hemorrhagic CCM lesion close to the pre-existing DVA. (**B**) Postoperative axial MRI demonstrating complete removal of the newly formed CCM lesion.

**Figure 3 ijms-23-14643-f003:**
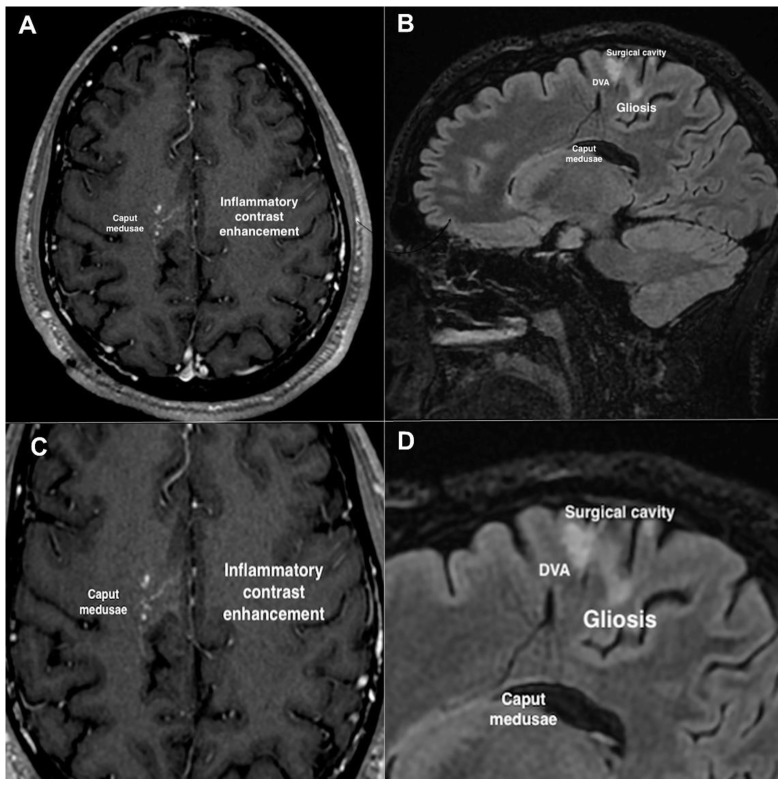
Contrast-enhanced MRI follow-up examination of the patient’s brain at 1 year after CCM surgery. (**A**,**C**) T1-weighted axial image. (**B**,**D**) T2-FLAIR (fluid-attenuated inversion recovery) sagittal image. Contrast-enhanced images were obtained after injection of gadolinium as nontoxic paramagnetic contrast enhancement agent. (**C**,**D**) Magnified images of panels (**A**,**B**), respectively. Notice the presence of signal-intensity abnormalities in the bed of the DVA, including point-like contrast enhancements and hyperintense gliotic areas.

**Figure 4 ijms-23-14643-f004:**
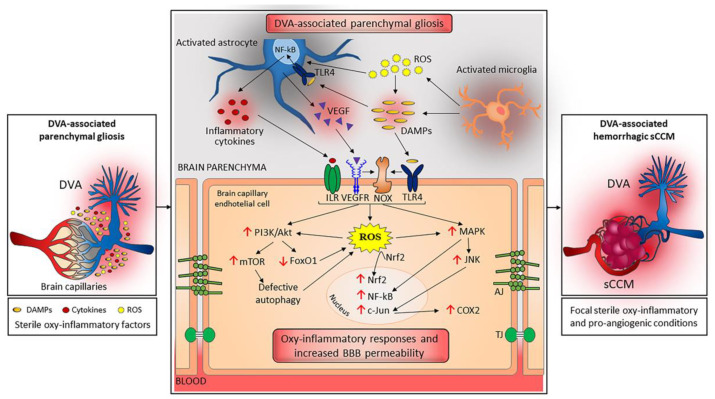
Graphical representation of focal sterile oxy-inflammatory mechanisms potentially implicated in the formation and recurrence of a sporadic cerebral cavernous malformation (sCCM) in the drainage territory of a developmental venous anomaly (DVA). Sterile oxy-inflammatory and pro-angiogenic factors, including DAMPs, inflammatory cytokines, ROS, and VEGF, can be released by damaged neurovascular tissues and activated glial cells as a consequence of age-related increased abnormalities in the angioarchitectural and hemodynamic features of a DVA, as well as by other injuries that can contribute to focal gliosis in the adjacent brain parenchyma, such as radiation exposure, neurodegeneration, stroke, and hypoxia. In turn, such factors can induce aberrant oxy-inflammatory signaling pathways in local brain capillary endothelial cells through the activation of multivalent platforms of distinct but interrelated cell surface receptors and enzymes, including interleukin receptors (ILR), VEGF receptors (VEGFR), Toll-like receptors (TLR4), and NADPH oxidases (NOX), which eventually result in abnormal oxy-inflammatory and proangiogenic responses, thereby affecting the BBB stability and permeability, and enabling the focal onset of CCM lesions. In particular, these effects are mediated by multiple redox-dependent mechanisms, including NOX-mediated ROS production, upregulation of PI3K/Akt and MAPK signaling, modulation of major redox-sensitive transcription factors, such as FoxO1, NF-κB, c-Jun, and Nrf2, and mTOR activation-dependent downregulation of autophagy. Eventually, these multiple and interconnected mechanisms lead to abnormal adaptive responses that impair microvessel barrier function and increase their sensitivity to local oxy-inflammatory insults. Notably, these pathological effects can be further facilitated by gain-of-function mutations in genes involved in the PI3K/Akt and MAPK pathways, which have indeed been identified in surgical specimens of sCCM lesions (see Section 3 for further details). Abbreviations: AJ, adherens junction; BBB, blood–brain barrier; COX-2, cyclooxygenase-2; DAMPs, damage-associated molecular patterns; FoxO1, forkhead box O1; ILR, interleukin receptor; JNK, c-Jun N-terminal kinase; MAPK, mitogen-activated protein kinase; mTOR, mechanistic target of rapamycin; NF-κB, nuclear factor-κB; NOX, NADPH oxidases; Nrf2, nuclear factor erythroid 2-related factor 2; PI3K, phosphoinositide 3-kinase; ROS, reactive oxygen species; TJ, tight junctions; TLR4, Toll-like receptor 4; VEGF, vascular endothelial growth factor; VEGFR, VEGF receptor.

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
