# Peer review of "Distant Recurrence of a Cerebral Cavernous Malformation in the Vicinity of a Developmental Venous Anomaly: Case Report of Local Oxy-Inflammatory Events"

_ijms, 2022, doi:10.3390/ijms232314643_

Round 1
Reviewer 1 Report
The authors described an interesting case of a cavernoma recurrence where inflammation was seen in the vicinity of the associated developmental venous anomaly. The pathophysiology proposed by the author is novel. However, there are some major concerns regarding the presentation of the findings and the rationale of the discussion:
Figure 3 - It looks like the oedema is around the surgical cavity rather than related to DVA, please zoom in the images to make it clearer. This is the key figure so it is important to be clear
Where is the PET data (para 1 pg 5)
Much of Discussion esp 3.1 can be consolidated into the Introduction (which needs shortening)
Discussion needs to be more related to the actual case rather than a literature summary
Para 2, pg 6 – “In our case DVA represents the main stressful event for recurrent CCM lesion formation and bleeding after neurosurgical removal.” This is a hypothesis – “may represent”
“Finally, DVA play an important role in CCM onset, progression, and clinical symptoms, when associated with the sporadic form of the disease.” – “may play”
Para 2, pg 8 – “Our data confirm(s)” – no it doesn’t
“Severe” should read “serious” – that reads better
The main criticism is that I am not convinced the results shown support the discussion at all. It just looks like there is oedema was around the surgical cavity, which happens to be next to the DVA. There may just be some small residual hidden that led to a rebleed. Please rewrite the article with significant changes
Minor points:
1) Abstract Methods: I will delete “rather particular”
2) Same for para 2, pg 3
3) Para 1, pg 4 – “On June 2018, following an acute onset of critical episodes characterized by paresthesia in the left lower limb with "walking" ascending to the hemifacial, followed by clones of the left arm, without disturbance of consciousness, the patient underwent a CT scan showing a right frontal parasagittal hemorrhagic lesion.” This sentence can be improved
a. Why is “walking” in quotations?
b. Ascending to “rest of hemibody”?
4) Please describe neurosurgical approach for the resection procedures
5) Para 2 Pg 4 – long TR MRI = FLAIR?
6) Figures – please specify what modalities are used eg T1+C, FLAIR etc
7) When was the post-op scans taken (Figure 3)?
8) Para 3, pg 7 – “arterio” rather than “artero”
Reviewer 2 Report
The authors presented a case report of recurrent ICH in the setting of CCM with an associated DVA. What would make this case so special compared to previously reported cases? "DVA induced inflammation" can not be concluded without any molecular or pathological analysis.
